# Ethylene-mediated nitric oxide depletion pre-adapts plants to hypoxia stress

Sjon Hartman [1], Zeguang Liu[1], Hans van Veen[1], Jorge Vicente[2], Emilie Reinen[1], Shanice Martopawiro[1], Hongtao Zhang [3], Nienke van Dongen[1], Femke Bosman[1], George W. Bassel[4], Eric J.W. Visser[5], Julia Bailey-Serres [1,6], Frederica L. Theodoulou[3], Kim H. Hebelstrup [7], Daniel J. Gibbs [4], Michael J. Holdsworth [2], Rashmi Sasidharan [1] & Laurentius A.C.J. Voesenek[1]

Timely perception of adverse environmental changes is critical for survival. Dynamic changes in gases are important cues for plants to sense environmental perturbations, such as submergence. In *Arabidopsis thaliana*, changes in oxygen and nitric oxide (NO) control the stability of ERFVII transcription factors. ERFVII proteolysis is regulated by the N-degron pathway and mediates adaptation to flooding-induced hypoxia. However, how plants detect and transduce early submergence signals remains elusive. Here we show that plants can rapidly detect submergence through passive ethylene entrapment and use this signal to pre-adapt to impending hypoxia. Ethylene can enhance ERFVII stability prior to hypoxia by increasing the NO-scavenger PHYTOGLOBIN1. This ethylene-mediated NO depletion and consequent ERFVII accumulation pre-adapts plants to survive subsequent hypoxia. Our results reveal the biological link between three gaseous signals for the regulation of flooding survival and identifies key regulatory targets for early stress perception that could be pivotal for developing flood-tolerant crops.

---

[1] Plant Ecophysiology, Institute of Environmental Biology, Utrecht University, Padualaan 8, 3584 CH Utrecht, The Netherlands. [2] School of Biosciences, University of Nottingham, Loughborough LE12 5RD, UK. [3] Plant Sciences Department, Rothamsted Research, Harpenden AL5 2JQ, UK. [4] School of Biosciences, University of Birmingham, Edgbaston B15 2TT, UK. [5] Department of Experimental Plant Ecology, Institute for Water and Wetland Research, Radboud University Nijmegen, 6525 AJ Nijmegen, the Netherlands. [6] Botany and Plant Sciences Department and Center for Plant Cell Biology, University of California, Riverside, CA 92521, USA. [7] Department of Molecular Biology and Genetics, Aarhus University, Forsøgsvej 1, DK-4200 Slagelse, Denmark. Correspondence and requests for materials should be addressed to M.J.H. (email: Michael.Holdsworth@nottingham.ac.uk) or to R.S. (email: R.sasidharan@uu.nl) or to L.A.C.J.V. (email: L.a.c.j.voesenek@uu.nl)

The increasing frequency of floods due to climate change[1] has devastating effects on agricultural productivity worldwide[2]. Due to restricted gas diffusion underwater, flooded plants experience cellular oxygen ($O_2$) deprivation (hypoxia) and survival strongly depends on molecular responses that enhance hypoxia tolerance[2,3]. In submerged plant tissues the limited gas diffusion causes passive ethylene accumulation. This rapid ethylene build-up can occur prior to the onset of severe hypoxia, making it a timely and reliable signal for submergence[4,5]. In several plant species, ethylene regulates adaptive responses to flooding by inducing morphological and anatomical modifications that prevent hypoxia[5]. Surprisingly, ethylene has so far not been linked to metabolic responses that reduce hypoxia damage. In addition, how plants perceive early submergence to subsequently increase survival remains elusive.

Here we show that plants can quickly sense submergence using passive ethylene accumulation and integrate this signal to acclimate to subsequent hypoxia. This ethylene-mediated hypoxia acclimation is dependent on enhanced group VII Ethylene Response Factor (ERFVII) stability prior to hypoxia. We show that ethylene limits ERFVII proteolysis under normoxic conditions by increasing the NO-scavenger PHYTOGLOBIN1 (PGB1). Our results reveal a molecular mechanism that plants use to integrate early stress signals to pre-adapt to forthcoming severe stress.

## Results

**Early ethylene signalling enhances hypoxia acclimation.** To unravel the spatial and temporal dynamics of ethylene signalling upon plant submergence, we monitored the nuclear accumulation of ETHYLENE INSENSITIVE 3 (EIN3)[6–9], an essential transcription factor for mediating ethylene responses. We show, through an increase in EIN3-GFP fluorescence signal, that ethylene is rapidly perceived (within 1–2 h) in *Arabidopsis thaliana* (hereafter Arabidopsis) root tips upon submergence (Supplementary Fig. 1a-c). An ethylene or submergence pre-treatment of only 4 h was sufficient to increase root meristem survival during subsequent hypoxia ( < 0.01% $O_2$). These responses were abolished in ethylene signalling mutants or via chemical inhibition of ethylene action (Supplementary Fig. 1d–e). Ethylene-induced acclimation to hypoxia was observed in both roots and shoots and was accompanied by a reduction in cellular damage in response to hypoxia (Fig. 1, Supplementary Figs. 2 & 3). Furthermore, enhanced hypoxia tolerance after ethylene pre-treatment is conserved within Arabidopsis accessions and taxonomically diverse flowering plant species, although variation in capacity to benefit from an ethylene pre-treatment exists (Supplementary Fig. 4;[10]). These results demonstrate that ethylene enhances tolerance of multiple plant organs and species to hypoxia. Next, we aimed to unravel how early ethylene signalling leads to enhanced hypoxia tolerance in Arabidopsis root tips.

**Ethylene stabilizes group VII Ethylene Response Factors.** Hypoxia acclimation in plants involves the up-regulation of hypoxia adaptive genes that control energy maintenance and oxidative stress homeostasis[11]. Interestingly, most of these genes were not induced by ethylene alone, but showed increased transcript abundance upon hypoxia following a pre-treatment with ethylene (Supplementary Fig. 5). Hypoxia adaptive genes are regulated by the ERFVII transcription factors that are components of a mechanism that senses $O_2$ and NO via the Cys-branch of the PROTEOLYSIS 6 (PRT6) N-degron pathway[12–14]. ERFVIIs are degraded following oxidation of amino terminal (Nt-) Cysteine in the presence of $O_2$ and NO, catalysed by PLANT CYSTEINE OXIDASEs (PCOs)[15]. The N-recognin E3 ligase

PRT6 promotes degradation of oxidized ERFVIIs by the 26 S proteasome[16,17]. A decline in either $O_2$ or NO stabilizes ERFVIIs, leading to transcriptional up-regulation of hypoxia adaptive genes and other environmental and developmental responses[12–14,18]. The constitutively synthesized ERVIIs RELATED TO APE-TALA2.12 (RAP2.12), RAP2.2 and RAP2.3 redundantly act as the principal activators of many hypoxia adaptive genes[19–21]. In contrast, HYPOXIA RESPONSIVE ERF1 (HRE1) and HRE2 function downstream of RAP-type ERFVIIs, being transcriptionally induced once hypoxia occurs[22]. We investigated whether ethylene-induced hypoxia tolerance depends on the constitutively synthesized RAP-type ERFVIIs. Single loss-of-function mutants of *RAP2.12*, *RAP2.2* and *RAP2.3*, and the *hre1 hre2* double mutant, responded to ethylene pre-treatment similarly to their WT backgrounds (Supplementary Fig. 6a). However, two independent *rap2.2 rap2.12* loss-of-function double mutants[20] showed no improved hypoxia tolerance after ethylene pre-treatment (Fig. 2a), while their WT background crosses did (Supplementary Fig. 6b). In contrast, overexpression of a stable N-terminal variant of RAP2.12[21], or inhibition of the PRT6 N-degron pathway in the *prt6–1* mutant[12,23] both enhanced hypoxia tolerance without an ethylene pre-treatment (Fig. 2a). These data indicate that ethylene-induced hypoxia tolerance occurs through the PRT6 N-degron pathway and redundantly involves at least RAP2.2 and RAP2.12[20,21].

We next explored how ethylene regulates *ERFVII* mRNA and protein abundance. Ethylene increased *RAP2.2*, *RAP2.3*, *HRE1* and *HRE2* transcripts in root tips and *RAP2.12*, *RAP2.2* and *RAP2.3* mRNAs in shoots (Supplementary Fig. 6c, d). Visualization and quantification of RAP2.12 abundance using transgenic *promRAP2.12:RAP2.12-GUS* and *35S:RAP2.12-GFP* protein-fusion lines revealed that ethylene strongly increased RAP2.12 protein in meristematic zones of main and lateral root tips and shoots under normoxia (Fig. 2b, c, Supplementary Fig. 6e, f). Since *35S:RAP2.12-GFP* is uncoupled from ethylene-triggered transcription, this suggests that ethylene limits ERFVII protein turnover. In root tips, this RAP2.12 stabilization appeared within nuclei across most cell types and was also independent of ethylene-enhanced *RAP2.12* transcript abundance (Fig. 2b, c, Supplementary Figure 6c, e, f). These data suggest that ethylene-enhanced ERFVII accumulation is regulated by post-translational processes.

**Ethylene limits ERFVII proteolysis through NO depletion.** To investigate enhanced ERFVII stability under ambient $O_2$, we studied the effect of ethylene on the expression of genes encoding PRT6 N-degron pathway enzymes or other mechanisms reported to influence ERFVII stability. In response to ethylene, none of these genes showed changes in transcript abundance (Supplementary Fig. 7a, b). In addition, as both $O_2$ and NO promote ERFVII proteolysis[17], and since ethylene was administered at ambient $O_2$ conditions (21%; normoxia) and did not lead to hypoxia in desiccators (Supplementary Fig. 7c), it is unlikely that hypoxia causes the observed ERFVII stabilization. Furthermore, while recent reports show that plants contain a hypoxic niche in shoot apical meristems and lateral root primordia[24,25], we did not observe enhanced hypoxia target gene expression in root tips exposed to ethylene treatments (Supplementary Fig. 5), ruling out ethylene-enhanced local hypoxia in these tissues.

Since NO was previously shown to control proteolysis of ERFVIIs and other $Met_1$-$Cys_2$ N-degron targets[14,18,26], we hypothesized that ethylene may regulate NO levels. Roots treated with the NO probe 4-Amino-5-Methylamino-2',7'-Difluorofluorescein (DAF-FM) Diacetate[27] revealed an ethylene-induced depletion in fluorescence, indicating that ethylene mediates NO

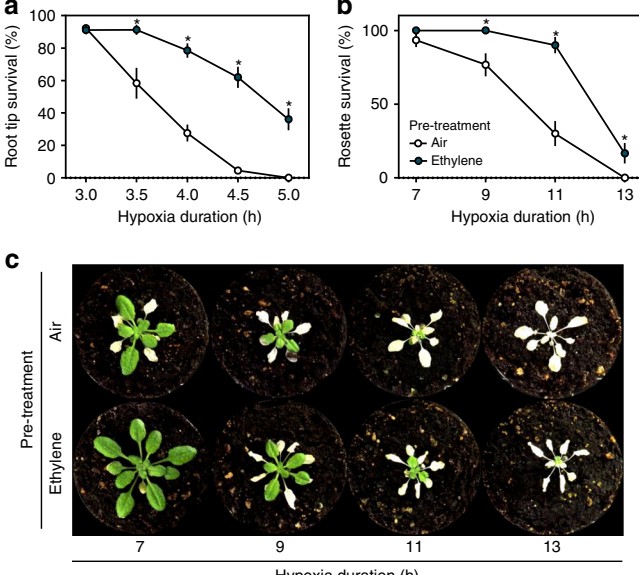

**Fig. 1** Ethylene pre-treatment enhances hypoxia tolerance. **a**, **b** Arabidopsis (Col-0) seedling root tip **a** and adult rosette **b** survival after 4 h of air (white) or ~5µll⁻¹ ethylene (blue) followed by hypoxia and recovery (3 days for root tips, 7 days for rosettes). Values are relative to control (normoxia) plants (mean ± sem). Asterisks indicate significant differences between air and ethylene ($p < 0.05$, Generalized linear model, negative binomial error structure, $n = 4$-8 rows consisting of ~23 seedlings **a**, $n = 30$ plants **b**). **c** Arabidopsis (Col-0) rosette phenotypes after 4 h of pre-treatment (air/ ~5µll⁻¹ ethylene) followed by hypoxia and 7 days recovery. All experiments were replicated at least 3 times

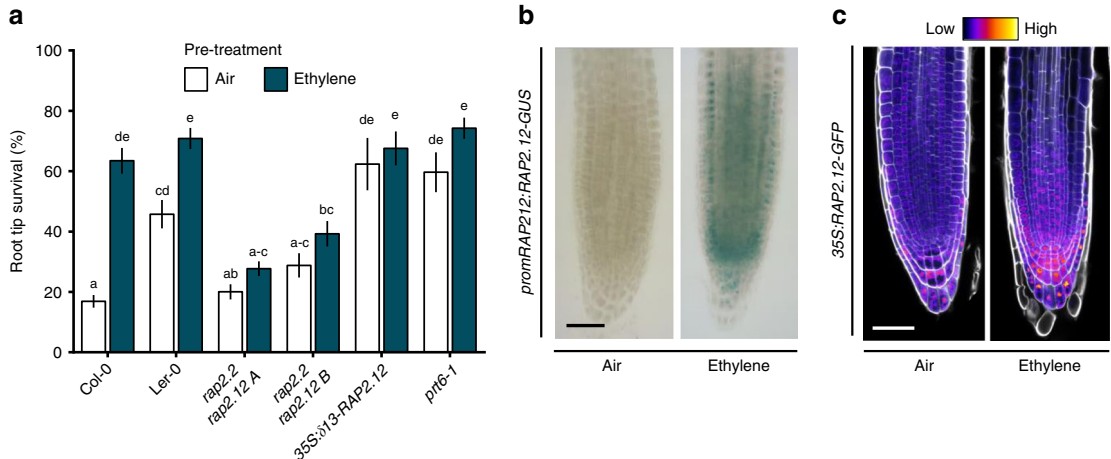

**Fig. 2** Ethylene-induced hypoxia tolerance is regulated by RAP-type ERFVIIs. **a** Seedling root tip survival of Col-0, Ler-0, *rap2.2 rap2.12* (2 independent lines in Col-0 x Ler-0 background), a constitutively expressed stable version of RAP2.12 and N-degron pathway mutant *prt6-1* after 4 h air or ~5µll⁻¹ ethylene followed by 4 h of hypoxia and 3 days recovery. Values are relative to control (normoxia) plants (mean ± sem). Statistically similar groups are indicated using the same letter ($p < 0.05$, 2-way ANOVA, Tukey's HSD, $n = 20$-28 rows consisting of ~23 seedlings). **b**, **c** Representative root tip images showing *promRAP2.12:RAP2.12-GUS* staining and confocal images of *35 S:RAP2.12-GFP* intensity in root tips after 4 h of air or ~5µll⁻¹ ethylene. Cell walls were visualized using Calcofluor White stain **c**. Scale bar of **b** and **c** is 50 µm. All experiments were replicated at least 3 times

levels (Fig. 3a, b). Next, we investigated whether this decline in NO was required for RAP-type ERFVII stabilization. Both ethylene and the NO-scavenging compound 2-(4-Carboxyphenyl)-4,4,5,5-Tetramethylimidazoline-1-oxyl-3-oxide (cPTIO) led to increased RAP2.12 and RAP2.3 stability under normoxia (Fig. 3c–e). However, the ethylene-mediated increase in RAP2.12 and RAP2.3 stability was abolished when an NO pulse was applied concomitantly confirming a role for NO depletion in ethylene-triggered ERFVII stabilization of both these RAPs during normoxia. Application of hypoxia after pre-treatments resulted in stabilization of RAP2.12 and RAP2.3, demonstrating that the plants were viable and the PRT6 N-degron pathway could still be

impaired (Fig. 3c–e, Supplementary Fig. 7d). These data together illustrate that both RAP2.12 and RAP2.3 depend on ethylene-mediated NO-depletion to promote their stability.

The functional consequences of ethylene-induced NO-dependent RAP2.12 stabilization for hypoxia acclimation were studied in a root meristem survival assay. Ethylene pre-treatment enhanced hypoxia survival, which was largely abolished by an NO pulse (Fig. 3f). Furthermore, pre-treatment with cPTIO to scavenge intracellular NO before hypoxia resulted in increased survival in the absence of ethylene. In genotypes lacking RAP2.12 and RAP2.2 or overexpressing a stable N-terminal variant of RAP2.12, neither ethylene nor NO manipulation had any effect

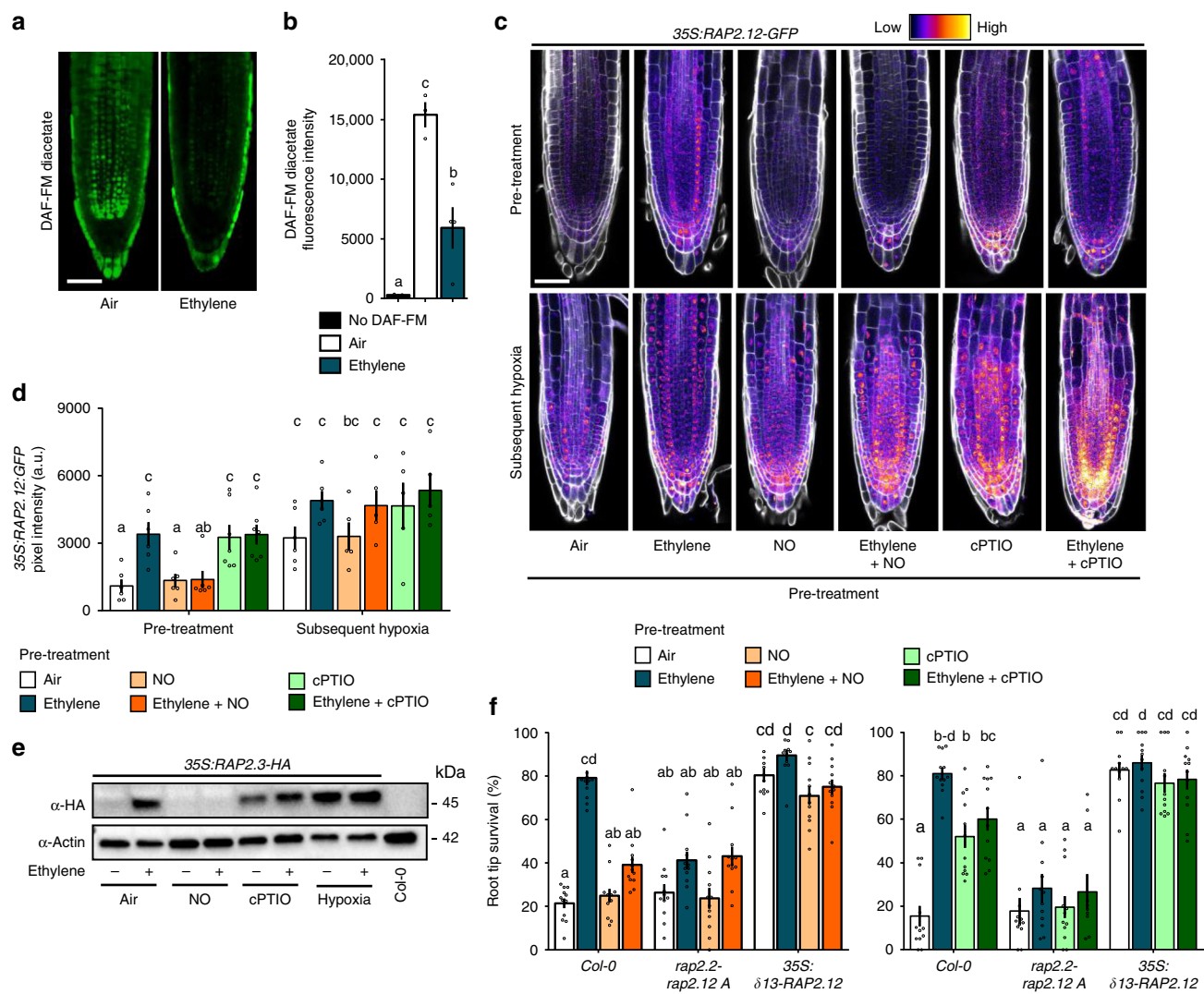

**Fig. 3** Ethylene impairs NO levels leading to ERFVII stability and enhanced hypoxia survival. **a**, **b** Representative confocal images visualizing **a** and quantifying **b** NO using fluorescent probe DAF-FM diacetate, in Col-0 seedling root tips after 4 h of air or ~5μll⁻¹ ethylene (scale bar= 50 μm). (Letters indicate significant differences (1-way ANOVA, Tukey's HSD, $n = 3$-4). **c**, **d** Representative confocal images visualizing **c** and quantifying **d** 35 S:RAP2.12-GFP intensity in seedling root tips after indicated pre-treatments and subsequent hypoxia (4 h). Cell walls were visualized using Calcofluor White stain (scale bar= 50μm). (Letters indicate significant differences ($p < 0.05$, 2-way ANOVA, Tukey's HSD, $n = 5$-7). **e** RAP2.3 protein levels in 35 S:MC-RAP2.3-HA seedlings (Col-0 background) after indicated treatments. **f** Seedling root tip survival of Col-0, rap2.2 rap2.12 line A mutants and an over-expressed stable version of RAP2.12 after indicated pre-treatments followed by hypoxia (4 h) and 3 days recovery. Values are relative to control (normoxia) plants. Letters indicate significant differences ($p < 0.05$, 2-way ANOVA, Tukey's HSD, $n = 12$ rows consisting of ~23 seedlings). All data shown are mean ± sem. All experiments were replicated at least 3 times, except for **c**, **d** and **f** (2 times)

on subsequent hypoxia survival (Fig. 3f). These results demonstrate that local NO removal, via cPTIO or as a result of elevated ethylene, is both essential and sufficient to enhance RAP2.12 and RAP2.3 stability during normoxia, and that increased hypoxia tolerance conferred by ethylene strongly depends on NO-mediated stabilization of RAP2.12 and RAP2.2 prior to hypoxia.

**The ethylene-mediated NO decline depends on PHYTOGLO-BIN1.** The question remained how ethylene regulates NO levels under normoxia. NO metabolism in Arabidopsis is mainly regulated by NO biosynthesis via NITRATE REDUCTASE (NR)-dependent nitrite reduction and NO-scavenging by three non-symbiotic phytoglobins (PGBs)[28–30]. Ethylene led to small increases in NR1 and NR2 mRNA levels, but this did not influence total NR activity (Supplementary Figure 8a, b, e). In contrast, transcript abundance of PGB1, the most potent NO-

scavenger[30], increased rapidly in root tips and shoots after ethylene treatment (Supplementary Fig. 8a–c). Importantly, PGB1 (a hypoxia-adaptive gene regulated by ERFVIIs) was still upregulated by ethylene during normoxia in rap2.2 rap2.12 mutant lines (Supplementary Fig. 8d). To study the effect of ethylene-induced PGB1 levels on NO metabolism, ERFVII stabilization, hypoxia-adaptive gene expression and hypoxia tolerance, we identified a T-DNA insertion line (SK_058388; hereafter pgb1–1). In pgb1–1 the T-DNA is located 300 bp upstream of the PGB1 start codon (Supplementary Fig. 9a, b). In wild-type plants, both ethylene and hypoxia treatment enhanced PGB1 transcript and protein accumulation (Fig. 4a, b). In pgb1–1, PGB1 transcript levels were reduced, and ethylene did not increase PGB1 transcript or protein abundance, whereas hypoxia only affected transcript abundance slightly (Fig. 4a, b). A faint band of lower molecular weight than expected for PGB1 (18 kDa) was observed in some pgb1–1 samples, but did not show any clear treatment

effect (Fig. 4b). Together these data illustrate that the T-DNA insertion in the promoter of *pgb1–1* uncouples *PGB1* expression from ethylene regulation. Conversely, a *35 S:PGB1* line had constitutively elevated *PGB1* transcript and protein levels (Fig. 4a, b[30],). Importantly, both *pgb1–1* and *35 S:PGB1* showed mostly similar ethylene responses in abundance of perception (*ETR2*) and biosynthesis (*ACO1*) transcripts compared to wild-type during normoxia (Supplementary Fig. 10), indicating that ethylene biosynthesis and signalling are unlikely to be affected.

The ethylene-mediated NO decline observed in wild-type root tips was fully abolished in *pgb1–1*, demonstrating the requirement of *PGB1* induction for local NO removal upon ethylene exposure (Fig. 4c, d). Moreover, lack of NO removal by ethylene in *pgb1–1* resulted in the inability to stabilize RAP2.3 levels and reduced hypoxia survival (Fig. 4e, f). These effects could be rescued by restoration of NO-scavenging capacity using cPTIO (Fig. 4f). In addition, the reduced ethylene-induced hypoxia tolerance in *pgb1-1* was also accompanied by an absence of enhanced hypoxia adaptive gene expression after an ethylene pre-treatment (Supplementary Fig. 10). In contrast, *35 S:PGB1* showed constitutively low NO levels in root tips (Fig. 4c, d[30],), and increased RAP2.3 stability under normoxia (Fig. 4e). Moreover, ectopic *PGB1* over-expression enhanced hypoxia tolerance without an ethylene pre-treatment, but this effect can be abolished by an NO pulse (Fig. 4f). Elevated mRNA levels for several hypoxia adaptive genes accompanied this constitutive hypoxia tolerance in *35 S: PGB1* root tips (Supplementary Fig. 10). These results demonstrate that active reduction of NO levels by ethylene-induced *PGB1* prior to hypoxia can precociously enhance ERFVII stability to prepare cells for impending hypoxia.

## Discussion

We show that plants have the remarkable ability to detect submergence quickly by passive ethylene entrapment and use this signal to acclimate to forthcoming hypoxic conditions. The early ethylene signal prevents N-degron targeted ERFVII proteolysis by increased production of the NO-scavenger PGB1 and in turn primes the plant's hypoxia response (Summarizing model, Fig. 5). Interestingly, while ethylene signalling prior to hypoxia leads to nuclear stabilization of RAP2.12 in root meristems (Figs. 2b, c, 3c), it does not trigger accumulation of most hypoxia adaptive gene transcripts until hypoxia occurs (Supplementary Fig. 5). Apparently, stabilization of ERFVIIs alone is insufficient to trigger full activation of hypoxia-regulated gene transcription and additional hypoxia-specific signals such as altered ATP and/or Ca$^{2+}$ levels are required[31–33]. The possible existence of undiscovered plant O$_2$ sensors was recently discussed and could potentially fulfil this role[34]. Furthermore, the current discovery of ethylene-mediated stability of ERFVIIs paves the way towards unravelling how ethylene could influence the function of the other recently discovered PRT6 N-degron pathway targets VERNALIZATION2 (VRN2) and LITTLE ZIPPER2 (ZPR2)[24,26].

This study shows that PGB1 is a key intermediate, linking ethylene signalling, via regulated NO removal, to O$_2$ sensing and hypoxia tolerance. This mechanism also provides a molecular explanation for the protective role of PGB1 during hypoxia and submergence described in prior studies[30,35–37]. Natural variation for ethylene-induced hypoxia adaptation was also observed in wild species and correlated with *PGB1* induction[10]. Our discovery provides an explanation for this natural variation and could be instrumental in enhancing conditional flooding tolerance in crops via manipulation of ethylene responsiveness of *PGB1* genes. In these modified plants, rapid passive ethylene entrapment upon flooding would increase PGB1 levels and pre-adapt crops to later occurring hypoxia stress.

## Methods

**Plant material**. Plant material: *Arabidopsis thaliana* seeds of ecotypes Col-0, Cvi-0, C24 and mutants *ein2-5* and *ein3eil1-1*[38,39] were obtained from the Nottingham Arabidopsis Stock Centre. Seeds of *pgb1-1* (SALK_058388) were obtained from the Arabidopsis Biological Resource Center and the molecular characterization of this line is described in Fig. 4a, b and Supplementary Fig. 9. Other germlines used in this study were kindly provided by the following individuals: Ler-0, *rap2.2-5* (Ler-0 background, AY201781/GT5336), *rap2.12-2* (SAIL_1215_H10), *rap2.2-5rap2.12-A* and *-B* (mixed Ler-0 and Col-0 background) from Prof. Angelika Mustroph[20], University Bayreuth, Germany; *35 S:δ13-RAP2.12-GFP* and *35 S:RAP2.12-GFP* from Prof. Francesco Licausi, University of Pisa, Italy;[13] and *35 S:EIN3-GFP* (*ein3eil1* mutant background) from Prof. Shi Xiao, Sun Yat-sen University, China[7]. The *35 S:PGB1*, *35 S:RAP2.3-HA* transgenic lines, as well as *prt6-1* (SAIL_1278_H11), *rap2.3-1* (SAIL_1031_D10) and *hre1-1hre2-1* (SALK_039484 + SALK_052858) mutants were described in the following publications by the authors of this study:[12,14,40]. Barley seeds were obtained from Flakkebjerg Research Center Seed Stock (Aarhus University). Additional mutant combinations used in this study were generated by crossing, and all lines were confirmed by either conventional genotyping PCRs and/or antibiotic resistance selection (Primer and additional info in Supplementary Table 1).

**Plant growth conditions**. Growth conditions adult rosettes: Arabidopsis seeds were placed on potting soil (Primasta) in medium sized pots and stratified at 4 °C in the dark for at least 3 days. Pots were then transferred to a growth chamber for germination under short day conditions (8:00 – 17:00, T = 20 °C, Photon Flux Density = ~150 μmol m$^{-2}$s$^{-1}$, RH = 70%). After 7 days, seedlings were transplanted individually into single pots (70 ml) that were filled with the similar potting soil (Primasta). Plants continued growing under identical short day conditions and were watered automatically to field capacity. Per genotype, homogeneous groups of 10-leaf-stage plants were selected and randomized over treatment groups for phenotypic and molecular analysis under various treatments. Plants used for hypoxia tolerance assays were transferred back to the same conditions after treatments to recover for 7 days.

Growth conditions seedlings: Seeds were vapor sterilized by incubation with a beaker containing a mixture of 50 ml bleach and 3 ml of fuming HCl in a gas tight desiccator jar for 3 to 4 h. Seeds were then individually transplanted in (2 or 3) rows of 23 seeds on sterile square petri dishes containing 25 ml autoclaved and solidified ¼ MS, 1% plant agar without additional sucrose. Petri dishes were sealed with gas-permeable tape (Leukopor, Duchefa) and stratified at 4 °C in the dark for 3 to 4 days. Seedlings were grown vertically on the agar plates under short day conditions (09:00–17:00 hours, T = 20 °C, Photon Flux Density = ~120 μmol m$^{-2}$ s$^{-1}$, RH = 70%) for 5 days for *Arabidopsis thaliana*, and 7 days for *Solanum lycopersicum* (Tomato, Moneymaker), *Solanum dulcamara* and *Arabidopsis lyrata* before phenotypic and/or molecular analysis under various treatments. For *Hordeum vulgare* (Barley, both ssp. Golden Promise and landrace Heimdal) seedlings were grown on agar in sterile tubs and were 3 days old before phenotypic analysis.

**Construction of transgenic plants**. The *promRAP2.12:MC-RAP2.12-GUS* protein fusion lines were constructed by amplifying the genomic sequence capturing 2 kb of sequence upstream of the translational start site, and removing the stop codon using the following primers: RAP2.12-fwd GGGGACAAGTTTGTAC AAAAAAGCAGGCTATTCAGATTGGATCGTGACATG and RAP2.12-rev GGGGACCACT TTGTACAAGAAAGCTGGGTAGAA- GACTCCTCCAATCATGGAAT. The PCR product was GATEWAY cloned into pDNR221 through a BP reaction, then transferred to pGWB433 creating an in-frame C-terminal fusion to the GUS reporter protein[41].

**Experimental setup and (pre-)treatments**. Ethylene treatments: Lids of the agar plates of the vertically grown seedlings were removed during all (pre-) treatments and plates were placed vertically into glass desiccators (22.5 L volume). Air (control) and ~5μll$^{-1}$ ethylene (pre-) treatments (by injection with a syringe) were applied at the start of the light period (09:00h for seedlings, 08:00h for adult rosettes) and were performed by keeping the seedlings/plants in air-tight closed glass desiccators under low light conditions (T = 20 °C, Light intensity = ~3-5 μmolm$^{-2}$ s$^{-1}$) for 4 h. Ethylene concentrations in all desiccators were verified by gas chromatography (Syntech Spectras GC955) at the beginning and end of the pre-treatment.

Hypoxia treatments: After 4 h of any pre-treatment plants/seedlings were flushed with oxygen-depleted air (humidified 100% N$_2$ gas) at a rate of 2.00 l/min under dark conditions to limit oxygen production by photosynthesis. Oxygen levels generally reached 0.00% oxygen within 40 min of the hypoxia treatment as measured with a Neofox oxygen probe (Ocean optics, Florida, USA) (Supplementary Fig. 7c). Control plants and seedlings were flushed with humidified air condition for the duration of the hypoxia treatment in the dark. Hypoxia treatment durations varied depending on the developmental stage and plant species and are specified in the appropriate figure legends.

Nitric oxide treatments: Just before application, pure NO gas was diluted in small glass vials with pure N$_2$ gas to minimize the oxidation of NO gas. Diluted NO

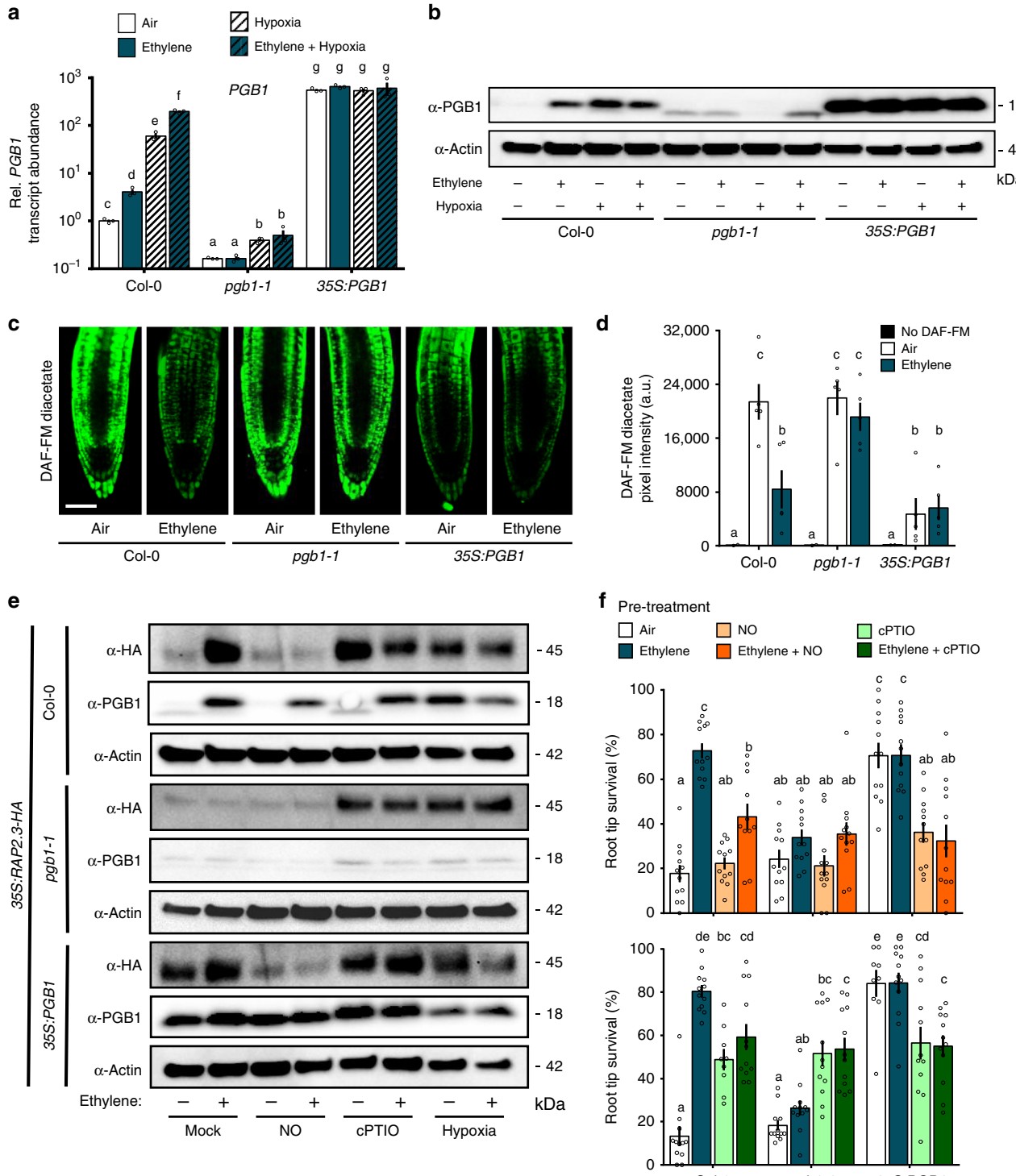

**Fig. 4** Ethylene mediates NO levels, ERFVII stability and hypoxia survival through PHYTOGLOBIN1. **a** Relative transcript abundance of *PGB1* in root tips of Col-0, *pgb1-1* and *35 S:PGB1* after 4 h air or ~5µll⁻¹ ethylene followed by (4 h) hypoxia. Values are relative to Col-0 air treated samples. Letters indicate significant differences (*p* < 0.05, 2-way ANOVA, *n* = 3 replicates of ~200 root tips each). **b** PGB1 protein levels in Col-0 , *pgb1-1* and *35 S:PGB1* root tips after 4 h air or ~5µll⁻¹ ethylene followed by (4 h) hypoxia. **c, d** Representative confocal images visualizing **c** and quantifying **d** NO using fluorescent probe DAF-FM diacetate in Col-0, *pgb1-1* and *35 S:PGB1* seedling root tips after 4 h air or ~5µll⁻¹ ethylene (scale bar= 50µm). Letters indicate significant differences (*p* < 0.05, 2-way ANOVA, Tukey's HSD, *n* = 5). **e** RAP2.3 and PGB1 protein levels in *35 S:MC-RAP2.3-HA* (in Col-0, *pgb1-1* and *35 S:PGB1* backgrounds) seedling root tips after indicated pre-treatments and subsequent hypoxia (4 h). **f** Seedling root tip survival of Col-0, *pgb1-1* and *35 S:PGB1* after indicated pre-treatments followed by 4 h hypoxia and 3 days recovery. Values are relative to control (normoxia) plants. Letters indicate significant differences (p < 0.05, 2-way ANOVA, Tukey's HSD, n = 12 rows of ~23 seedlings). All data shown are mean ± sem. All experiments were replicated at least 2 times

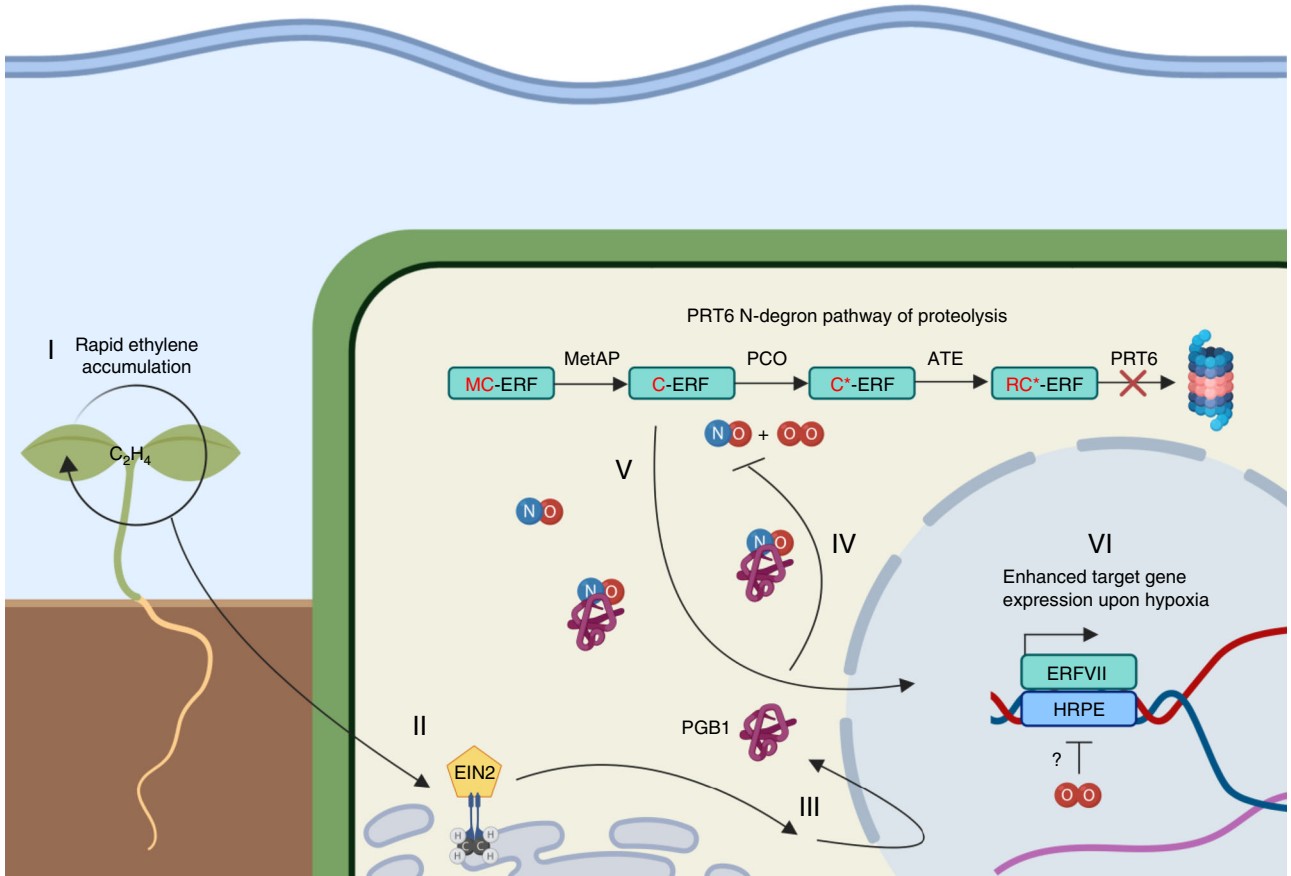

**Fig. 5** Proposed mechanism of ethylene-induced hypoxia tolerance upon submergence. **I** Upon submergence ethylene ($C_2H_4$) accumulates within minutes in plant tissues due to restricted gas diffusion. **II+III** Ethylene perception leads to EIN2 and EIN3EIL1 dependent signalling and enhanced production of NO-scavenger PHYTOGLOBIN1 (PGB1) within 1 h of ethylene signalling. **IV+V** Within 4 h, these enhanced PGB1 levels lead to NO depletion, in turn limiting PRT6 N-degron pathway targeted proteolysis of RAP-type group VII Ethylene Response Factor transcription factors (ERVIIs). **V+VII** Stabilized ERFVIIs translocate to the nucleus where they induce enhanced hypoxia gene expression only when $O_2$ deprivation occurs. This amplified hypoxia response increases hypoxia tolerance of Arabidopsis root and shoot apical meristems (created with BioRender.com)

gas was injected with a syringe into the air and ethylene treated desiccators at a final concentration of 10 ull$^{-1}$ NO, 1 h prior to the end of the (pre-)treatment.

cPTIO applications: Treatments with the NO-scavenger 2-(4-Carboxyphenyl)-4,5-dihydro-4,4,5,5-tetramethyl-1H-imidazol-1-yloxy-3-oxide potassium salt (cPTIO salt, Sigma Aldrich, Darmstadt, Germany) were performed 1 h prior to air/ethylene treatments to allow for treatment combinations. Droplets of 5 µl cPTIO solution (250 µM in autoclaved liquid ¼ MS) or mock solution (autoclaved liquid ¼ MS) were pipetted onto each individual root tip.

1-MCP treatments: Seedlings were placed in closed glass desiccators (22.5 l volume) and gassed with 5µll$^{-1}$ 1-MCP (Rohmand Haas) for 1 h prior to other (pre-) treatments.

Submergence treatments: For submergence (pre-) treatments, the plates of vertically grown seedlings were placed horizontally and were carefully filled with autoclaved tap water until the seedlings were fully submerged.

**Hypoxia tolerance assays.** Survival of adult rosette plants: 10-leaf stage plants received ethylene and air pre-treatments followed by several durations of hypoxia and were subsequently placed back under short day growth chamber conditions to recover. After 7 days of recovery survival rates and biomass (fresh and dry weight) of surviving plants were determined.

Root tip survival of seedlings: 5-day old seedlings grown vertically on agar plates received pre-treatments (described above) followed by several durations of hypoxia (generally 4 h for mutant analysis). After the hypoxia treatment, agar plates were closed and sealed again with Leukopor tape and the location of root tips was marked at the back of the agar plate using a marker pen (0.8 mm fine tip). Plates were then placed back vertically under short day growth conditions for recovery. After 3-4 days of recovery, seedling root tips were scored as either alive or dead based on clear root tip re-growth beyond the line on the back of the agar plate. Primary root tip survival was calculated as the percentage of seedlings that showed root tip re-growth out of a row of (maximally) 23 seedlings. Root tip survival was expressed as relative survival compared to control plates that received similar pre-treatments but no hypoxia. For *Solanum lycopersicum* (Tomato, Moneymaker),

*Solanum dulcarama* and *Arabidopsis lyrata* methods were similar as described above, but seedlings were 7 days old. For *Hordeum vulgare* (Barley, both ssp. Golden Promise and landrace Heimdal) seedlings were only 3 days old and received 20 h of hypoxia before scoring survival of whole seedlings after 3 days of recovery.

**Evans blue staining for cell viability in root tips.** Arabidopsis seedlings were taken for root cell integrity analysis by Evans blue staining after air and ethylene pre-treatments followed by both hypoxia and post-hypoxia time-points. Seedlings were incubated in 0.25% aqueous Evans blue staining solution for 15 min in the dark, subsequently washed three times with Milli-Q water to remove excess dye and finally imaged using light microscopy (OLYMPUS BX50WI, 10x objective). Evans blue area and pixel intensity of the microscopy images was analyzed using ICY software (http://icy.bioimageanalysis.org/), by quantifying the mean pixel intensity of the red (ch0) and blue (ch2) channels of the tissues of interest, and expressed as Blue/Red pixel intensity.

**RNA and RT-qPCR.** Adult rosette (2 whole rosettes per sample), whole seedling (~20 whole seedlings) or seedling root tip (~200-500 root tips) samples were harvested by snap freezing in liquid nitrogen. Total sample RNA was extracted from frozen pulverized tissue using the RNeasy Plant Mini Kit protocol (Qiagen, Dusseldorf, Germany) with on-column DNAse treatment Kit (Qiagen, Dusseldorf, Germany) and quantified using a NanoDrop ND-1000 UV-Vis spectrophotometer (Nanodrop Technology). Single-stranded cDNA was synthesized from 500 ng RNA using random hexamer primers (Invitrogen, Waltham, USA). RT-qPCR was performed using the Applied Biosystems ViiA 7 Real-Time PCR System (Thermo Fisher Scientific) with a 5 µl reaction mixture containing 2.5 µl 2× SYBR Green MasterMix (Bio-Rad, Hercules, USA), 0.25 µL of both 10 µM forward and reverse primers and 2 µl cDNA (5 ng/µl). Average sample CT values were derived from 2 technical replicates. Relative transcript abundance was calculated using the comparative CT method[42], fold change was generally expressed as fold change relative

to air treated samples of Col-0. *ADENINE PHOSPHORIBOSYL TRANSFERASE 1* (*APT1*) was amplified, stable in all treatments and used as a reference gene. Primers used for RT-qPCR are listed in Supplementary Table 2.

**Histochemical staining for GUS activity**. Seedlings of *promRAP2.12:RAP2.12-GUS* (10 days old) were harvested in GUS solution (100 mM NaPO4 buffer, pH 7.0, 10 mM EDTA, 2 mM X-Gluc, 500 µM K3Fe(CN)6 and 500 µM K4Fe(CN)6) directly after (indicated in figure legend) treatments, vacuum infiltrated for 15 min and incubated for 2 days at 37 °C before de-staining with 70% ethanol. Seedlings were kept and mounted in 50% glycerol and analyzed using a Zeiss Axioskop2 DIC (differential interference contrast) microscope (10× DIC objective) or regular light microscope with a Lumenera Infinity 1 camera. GUS pixel intensity of the microscopy images was analyzed using ICY software (http://icy.bioimageanalysis.org/), by quantifying the pixel intensity of the red (ch0) and blue (ch2) channels of these images relative to the respective channel background values of these images. GUS intensity of all treatments was expressed relatively to the Air-treated controls.

**Protein extraction and Western Blotting**. Protein was extracted on ice for 30 min from pulverized snap frozen samples in modified RIPA lysis buffer containing 50 mM HEPES-KOH (pH 7.8), 100 mM KCl, 5 mM EDTA (pH 8), 5 mM EGTA (pH 8), 50 mM NaF, 10% (v/v) glycerol, 1% (v/v) IGEPAL, 0.5% (w/v) deoxycholate, 0.1% (w/v) SDS, 1 mM Na3VO4 (sodium orthovanadate), 1 mM PMSF, 1x proteinase inhibitor cocktail (Roche), 1x PhosSTOP Phosphatase Inhibitor Cocktail (Roche) and 50 µM MG132[43]. Protein concentration was quantified using and following the protocol of a BCA protein assay kit (Pierce). Protein concentrations were equalized by dilution with RIPA buffer and incubated for 10 min with loading buffer (5x sample loading buffer, Bio Rad) + β-ME) at 70 °C before loading (30 µg total protein per sample) on pre-cast Mini-PROTEAN Stain Free TGX Gels (Bio Rad) and ran by SDS-PAGE. Gels were imaged before and after transferring to PVDF membranes (Bio Rad) using trans-blot turbo transfer system (Bio Rad), to verify successful and equal protein transfer. Blots were blocked for at least 1 h in blocking solution at RT (5% milk in 1xTBS) before probing with primary antibody in blocking solution (α-HA-HRP, 1:2500 (Roche, Cat. No. 12 013 819 001)); α-PGB1, 1:500 (produced for this study using full length protein as antigen by GenScript); α-Actin, 1:2500 (Thermo Fisher Scientific, Cat. No. MA1-744) overnight at 4 °C. Blots were rinsed 3 times with 1xTBS-T (0.1% Tween 20) for 10 min under gentle agitation before probing with secondary antibody (α-rabbit IgG-HRP, Cat. No. 7074, for PGB1, 1:3000; α-mouse IgG-HRP, Cat. No. 7076, for Actin, 1:2500) and/or SuperSignal™ West Femto chemiluminescence substrate (Fisher Scientific) and blot imaging using Image Lab software in a chemi-gel doc (Bio-rad) with custom accumulation sensitivity settings for optimal contrast between band detection and background signal. To visualize RAP2.3 (~45 kDa) and ACTIN (~42 kDa) protein levels on the same blot, membranes were stripped after taking final blot images using a mild stripping buffer (pH 2,2, 1.5% (w/v) glycine, 0.1% SDS and 1.0% Tween 20) for 15 min and rinsed 3x in 1xTBS-T before blocking and probing with the 2nd primary antibody of interest.

**NO quantification**. Intracellular NO levels were visualized using DAF-FM diacetate (7'-difluorofluorescein diacetate, Bio-Connect). Seedlings were incubated in the dark for 15 min under gentle agitation in 10 mM Tris-HCl buffer (pH 7.4) containing 50 µM DAF-FM DA and subsequently washed twice for 5 min 10 mM Tris-HCl buffer (pH 7.4). Several roots of all treatments/genotypes were mounted in 10 mM Tris-HCl buffer (pH 7.4) on the same microscope slide. Fluorescence was visualized using a Zeiss Observer Z1 LSM700 confocal microscope (oil immersion, 40x objective Plan-Neofluar N.A. 1.30) with excitation at 488 nm and emission at 490–555 nm. Roots incubated and mounted in 10 mM Tris-HCl buffer (pH 7.4) without DAF-FM DA were used to set background values where no fluorescence was detected. Within experiments, laser power, pinhole, digital gain and detector offset were identical for all samples. Mean DAF-FM DA fluorescence pixel intensity in root tips was determined in similar areas of ~17,000 µm$^2$ between epidermis layers using ICY software (http://icy.bioimageanalysis.org/).

**Confocal Microscopy**. Transgenic Arabidopsis seedlings of *35 S:EIN3-GFP* and *35 S:RAP2.12-GFP* and were fixed in 4% PFA (pH 6.9) right after treatments, kept under gentle agitation for 1 h, were subsequently washed for 1 min in 1x PBS and stored in ClearSee clearing solution (xylitol 10% (w/v), sodium deoxycholate 15% (w/v) and urea 25% (w/v)[44]. Seedlings were transferred to 0.01% Calcofluor White (in ClearSee solution) 24 h before imaging. Fluorescence was visualized using a Zeiss Observer Z1 LSM700 confocal microscope (oil immersion, ×40 objective Plan-Neofluar N.A. 1.30) with excitation at 488 nm and emission at 490–555 nm for GFP and excitation at 405 nm and emission at 400–490 nm for Calcofluor White. Within experiments, laser power, pinhole, digital gain and detector offset were identical for all samples. Mean GFP fluorescence pixel intensity in root tips was determined in similar areas of ~17,000 µm$^2$ between epidermis layers using ICY software (http://icy.bioimageanalysis.org/).

**Nitrate reductase activity assay**. The NR activity was assessed using a mix of 20 whole 10-day-old seedlings with 2 replicates per treatment. Snap frozen samples were ground and homogenized in extraction buffer (100 mM HEPES (pH7.5), 2

mM EDTA, 2 mM di-thiothreitol, 1% PVPP). After centrifugation at 30,000 × g at 4 C for 20 min, supernatants were collected and added to the reaction buffer (100 mM HEPES (pH7.5), 100 mM NaNO3, 10 mM Cysteine, 2 mM NADH and 2 mM EDTA). The reaction was stopped by the addition of 500 mM zinc acetate after incubation for 15 min at 25 °C. Total nitrite accumulation was determined following addition of 1% sulfanilamide in 1.5 M HCl and 0.02% naphthylethylene-diamine dihydrochloride (NNEDA) in 0.2 M HCl by measuring the absorbance of the reaction mixture at 540 nm.

**Statistical analyses**. No statistical methods were used to predetermine sample size. Samples were taken from independent biological replicates. In general, the sample size of experiments was maximized and dependent on technical, space and/or time limitations. For root tip survival assay, the maximum amount of seedlings used per biological replicate, generally 1 row of seedlings for in vitro agar plates, is mentioned in the appropriate figure legends. Data was plotted using Graphpad Prism software. The statistical tests were performed two-sided, using either Graphpad Prism or R software and the "LSmeans and "multmultcompView" packages. Survival data was analyzed with either a generalized linear modeling (GLM) approach or an ANOVA on arcsin transformation of the surviving fraction. A negative binomial error structure was used for the GLM. Arcsin transformation ensured a homogeneity and normal distribution of the variances, especially for data that did not have treatments with all living or all death responses. The remaining data were analyzed with either Students t-test, 1-way or 2-way ANOVAs. Here data were log transformed if necessary to adhere to ANOVA prerequisites. Multiple comparisons were corrected for with Tukey's HSD.

**Reporting Summary**. Further information on research design is available in the Nature Research Reporting Summary linked to this article.

## Data availability

No restrictions are placed on materials and data availability. Biological materials such as mutant/transgenic lines can be requested from the corresponding authors. Details of all data and materials used in the analysis are available in the main text or the supplementary information. Gene accession numbers of all the Arabidopsis genes/mutants used in this study are listed in the Materials & Methods section and Supplementary Table 1 and 2. Source Data (including uncropped blots) related to Fig. 1a, b; Fig. 2a; Fig. 3b–f; Fig. 4a–f; Supplementary Fig. 1,b–e; Supplementary Fig. 2a, b; Supplementary Fig. 3b, c; Supplementary Fig. 4a–c; Supplementary Fig. 5; Supplementary Fig. 6a–d, f; Supplementary Fig. 7a–d; Supplementary Fig. 8a–e; and Supplementary Fig. 10 are provided with the paper.

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

## Acknowledgements

We thank the following individuals for providing seeds of these genotypes: Angelika Mustroph for Ler-0, Col-0 x Ler-0 WT crosses, *rap2.2*, *rap2.12* and *rap2.2 rap2.12-A & B*, Francesco Licausi for *35 S:δ13-RAP2.12-GFP* and *35 S:RAP2.12-GFP*, Shi Xiao for *35 S: EIN3-GFP* and Frank Becker for *Arabidopsis lyrata* seeds. We acknowledge Sophie Berckhan, Ankie Ammerlaan, Rob Welschen, Tamara Le Thanh, Johanna Kociemba, Florian de Deugd and Joris te Riele for technical assistance. Finally, we thank Ronald Pierik for feedback on the manuscript and Kasper van Gelderen and Jesse Küpers for their input on confocal imaging. This work was supported by grants from the Netherlands Organization for Scientific Research (831.15.001 to S.H., 824.14.007 to L.A.C.J.V, S. M. and BB.00534.1 to R.S.) and the Biotechnology and Biological Sciences Research Council [BB/M007820/1 and BB/K000144/1] to M.J.H.

## Author contributions

S.H, Z.L, H.v.v, J.V, H.Z, E.J.W.V, J.B.S, F.L.T, K.H.H, D.J.G, M.J.H, R.S. and L.A.C.J.V. designed experiments; S.H, Z.L, J.V.C, E.R, S.M, H.Z, N.v.d, F.B, G.W.B and E.J.W.V performed experiments; S.H, M.J.H, R.S. and L.A.C.J.V. wrote the manuscript.

## Additional information

**Competing interests:** The authors declare no competing interests.

9