## [Peer Review File · Nature Communications]

Reviewers' comments:

Reviewer #1 (Remarks to the Author):

Hartman et al. describe the important role of ethylene in triggering an early hypoxia response, which may allow the plants to pre-adapt to the stress. The authors also describe the role of PHYTOGLOBIN1 (PGB1), a scavenger of nitric oxide (NO), in mediating this role of ethylene in the onset of an adaptive hypoxia response. The results are overall very clearly and carefully presented and the Extended Data provides important evidence for the main points.

Important conclusions drawn in this paper rely on the pre-treatment of plants with ethylene in glass desiccators. However, oxygen levels at the end of the experiment have not been measured to ensure that they have not been affected by the ethylene treatment and/or plant responses to ethylene. Considering the strong overlap between hypoxia response and the observed effects in the presence of ethylene, verifying that the experimental set up has not resulted in lower oxygen levels in the course of the experiment appears to be an important control.

Figure 2: The authors use two rap2.2 rap2.12 double mutant lines isolated after crossing the single mutants with each other. The use of two lines serves as a control for the mixed L-er/Col-0 genetic background that results from the cross, however, considering the importance of this data for the paper, it would be good to either show the same result using alleles that are all in the Col-0 background (for example, previously published rap2.2 rap2.3 rap2.12 mutant lines), or using a couple of 'wild-type' controls re-isolated from the cross, and which should also have a mixed L-er/Col-0 background.

In general, the authors indicate the number of plants/seedlings used, which is very good, but they do not specify how many independent experiments (i.e. replicates) were conducted. This information should be included in the figure legends for all experiments carried out. Otherwise, it is difficult to assess the reproducibility. When 'n' is specified, it is unclear if the authors are referring to number of plants/seedlings or number of independent experiments. This needs clarification.

Minor suggestions:

Figure 4: although no mention of a potential role of RAP2.3 is made, and this has not been tested using genetic approaches, the authors use RAP2.3 in panel 4e to make their point for a connection between PGB1 overexpression, RAP2 levels and pre-adaptation to hypoxia. If similar blots could be conducted using RAP2.12 for example, this would strengthen the data further and would correlate with the genetic analysis at the beginning of the article.

Letters for figure panels are incorrect in some figure legends or on some figures (Figures 2 and 4).

Figure 4b: can the authors comment on the presence of a lower molecular weight protein recognized by the anti-PGB1 antibodies in the pgb-1 mutant background?

Reviewer #2 (Remarks to the Author):

In this paper, the authors report that the gaseous phytohormone ethylene can upregulate PHYTOGLOBIN-1 (PGB1), a nitric oxide (NO)-scavenger. The PGB1-mediated NO reduction stabilizes a transcription factor called Group VII ETHYLENE RESPONSE FACTOR (ERFVII). Thus, when a plant first detects flooding, ethylene-triggered NO depletion and the subsequent increase in ERFVII protein help it to prepare for the hypoxic stress that is expected to follow. This study provides a nice example of how the adaptation of plants to flooding stress involves the coordinated signaling

of three gases (O₂, ethylene and NO). Understanding the molecular mechanisms involved in this signaling is essential for the breeding of flood-tolerant crops. Abiotic stresses associated with climate change are likely to affect crop production. Thus, the mechanisms by which plants respond and adapt to abiotic stresses are of great interest. This study should therefore be of interest to many readers of Nature Communications.

I suggest that the authors consider the following points.

L50-52 (and Extended Data Fig. 1a, b & c): The authors evaluated how fast ethylene was perceived in root tips upon submergence, based on the intensities of fluorescence of the EIN3-GFP fusion protein. The authors need to explain how an increase in the fluorescence of EIN3-GFP is an indication that the root tips have perceived ethylene. Relevant papers should be cited for this.

L56: "S2 & S3" should be "Extended Data Figs. 2 & 3".

L84-85 (and Fig. 2a): The authors state that ethylene-induced hypoxia tolerance redundantly involves RAP-type ERFVIs, which implies that RAP2.3 shares this redundancy. However, Fig. 2a shows that the double mutants rap2.2/rap2.12 (in which RAP2.3 is still functional) do not have improved hypoxia tolerance after ethylene treatment. This seems to say that that RAP2.3 is not functionally redundant with RAP2.2 and RAP2.12 regarding hypoxia tolerance. The authors should explain their functions in detail.

L95: "Fig. S6b" should be "Extended Data Fig. 6b".

Fig. 2c: label "b" should be label "c".

Fig. 3d and 3f: The ethylene and ethylene+NO pre-treatments had similar effects on the intensities of RAP2.12::GFP fluorescence under subsequent hypoxia (Fig. 3d), but they had different effects on the rates of root tip survivals in Col-0 (Fig. 3f). A discussion about the different effects is needed.

Fig. 3e and Fig. 4e: The authors should measure RAP2.3-HA protein levels to examine the effects of NO and ethylene+NO pre-treatments by subsequent hypoxia treatment on accumulations of RAP2.3-HA protein. In this way, we can understand that the ethylene-mediated RAP2.3 stabilities are not abolished under subsequent hypoxia even if NO is pre-treated with ethylene, as like the case of RAP2.12::GFP fluorescence shown in Fig. 3d.

Extended Data Fig. 3: It is difficult to compare the cell damage in root tips between the air and ethylene pre-treatments. Can the authors quantify the Evans blue-stained colors?

Extended Data Fig. 4: There seems to be no common thread to the three panels (panel a, b and c show rosette FW, root tip survival and seedling survival, respectively). Is this figure needed?

Reviewer #3 (Remarks to the Author):

The manuscript (MS) submitted by Hartman et al. studies the cross road between gaseous molecules, and their regulators, that are known to be involved in plant responses to hypoxia stress. This new approach was intended to give answers on the participation of Ethylene (Ethy), nitric oxide (NO) and oxygen (O₂) in the regulation processes involved in the program that plants activate by low oxygen availability. Authors have studied the molecular mechanisms underpinning the adaptive process to hypoxia by which plants pretreated by low oxygen conditions are better prepared to respond to a subsequent hypoxic stress. In brief, they have mimicked hypoxia by pre-

treating plants with Ethy, NO or low oxygen and studied how plants survive to a next and more severe situation of low availability of oxygen like flooding condition. These are classical priming experiments. Authors go deeper in the understanding of plant responses to hypoxia trying to decipher the link between gases and some of their regulators. Knowing signaling pathways in responses to flooding stress could contribute to the design crop plants more tolerant to the consequences of the climate change.

Response to hypoxia is characterized by the stability of Group VII ethylene response factor (ERFVII), due to hypoxia-induced inhibition of the N-degron pathway, which is mediated by Ethy and NO. Oxygen and NO oxidizes and destabilizes ERFVII and prevent responses to hypoxia. Hypoxia-induced Phytoglobins expression depletes increases of the NO concentration during hypoxia. Authors use mutants in the N-degron pathway, ERFVII factor, Phytoglobins (know out mutants and overexpressed) to demonstrate the linkage between the gaseous molecules in the signaling pathway.

The experiments keep rationale and results are consistent with the statements. The novelty of results is barely at the standard level required by the journal.

The main points authors may address to improve the MS are the following.

1.- Title should be more specific. For instance, when authors say 'Gas signalling', clearly name the involved gaseous molecules that have been studied. Readers should clearly comprehend the statement. Do all studied gases allow plants pre-adapt to survive hypoxia stress? Which are those gases? In addition, title should clearly and specifically highlight the novelty of the attained results.
2.- It has been recently described that a hypoxic niche is required to set the activity of meristem cells through the regulation of the N-degron pathways, thus controlling the activity of stem cells (Weits et al., Nature, May-2019). In that report, authors conclude that hypoxic conditions were studied not as limiting factor, but rather as a condition in shoot apical meristem cells to promote leaf organogenesis.

Given that in the submitted MS, authors Hartman et al. used the root tip survival (where resides the root tip meristem) and root growth as a read out of plant response to hypoxic stress; it would be nice to integrate and discuss these results with those published in the recent Nature article, for instance integrating the roles of the presence or absence of oxygen in terms of both a signaling molecule and a critical substrate for cell metabolic activity.

Stem cells in root tip should keep in a hypoxic niche to commit their function, Are Ethy, NO and phytoglobins present in root meristem and playing there a different function than they do on differentiated cells?, Are there different signaling pathways responding to hypoxia in the stem cells of meristem compared to other cells? How authors analyze the priming treatments with gas and their effects, when integrating all responses of target cells considering their dissimilar intrinsic characteristics?

Reviewers' comments:

We thank all reviewers for their very constructive feedback and kind words. We have adjusted the manuscript to accommodate all issues raised by the reviewers and the details of this are explained below:

Reviewer #1 (Remarks to the Author):

Hartman et al. describe the important role of ethylene in triggering an early hypoxia response, which may allow the plants to pre-adapt to the stress. The authors also describe the role of PHYTOGLOBIN1 (PGB1), a scavenger of nitric oxide (NO), in mediating this role of ethylene in the onset of an adaptive hypoxia response. The results are overall very clearly and carefully presented and the Extended Data provides important evidence for the main points.

Important conclusions drawn in this paper rely on the pre-treatment of plants with ethylene in glass desiccators. However, oxygen levels at the end of the experiment have not been measured to ensure that they have not been affected by the ethylene treatment and/or plant responses to ethylene. Considering the strong overlap between hypoxia response and the observed effects in the presence of ethylene, verifying that the experimental set up has not resulted in lower oxygen levels in the course of the experiment appears to be an important control.

Response: We agree with the reviewer that this is an important control. To verify that ethylene does not modulate oxygen levels we monitored oxygen dynamics in our experimental set up. Measurements were done in the outflow of the desiccator during the ethylene pre-treatment and the start of the subsequent hypoxia phase. The experiment, which was repeated independently several times, showed consistently that oxygen levels within desiccators remained unaffected during the ethylene pre-treatment and only dropped upon the switch to N₂ treatment. The results are shown in Supplementary Figure 7c and is also referred to in the text (Line 120-123, page 5).

Figure 2: The authors use two rap2.2 rap2.12 double mutant lines isolated after crossing the single mutants with each other. The use of two lines serves as a control for the mixed L-er/Col-0 genetic background that results from the cross, however, considering the importance of this data for the paper, it would be good to either show the same result using alleles that are all in the Col-0 background (for example, previously published rap2.2 rap2.3 rap2.12 mutant lines), **or** using a couple of 'wild-type' controls re-isolated from the cross, and which should also have a mixed L-er/Col-0 background.

Response: We agree that including these wild-type controls would strengthen the main message. We were unable to generate a verified full knock-out line for all three RAPs (all available lines in Col-0 are leaky for RAP2.2, with a full knock-out being potentially lethal <https://www.ncbi.nlm.nih.gov/pmc/articles/PMC2048778/> and could therefore not be used). We therefore performed the alternative experiment suggested by the reviewer (Supplementary Figure 6b). We included 2 independent 'wild-type' controls isolated from the cross, which have a mixed Ler/Col-0 background, also published by Gasch, *et al.*, *Plant Cell*, 2015). We have also toned down our statements in the text regarding the clear involvement of RAP2.3 for mediating enhanced tolerance (line 101-102, page 4; line 150-152, page 6).

In general, the authors indicate the number of plants/seedlings used, which is very good, but they do not specify how many independent experiments (i.e. replicates) were conducted. This information should be included in the figure legends for all experiments carried out. Otherwise, it is difficult to assess the reproducibility. When 'n' is specified, it is unclear if the authors are referring to number of plants/seedlings or number of independent experiments. This needs clarification.

Response: We have now specified sample size (n), amount of pseudo-replicates within a sample (for instance, ~23 for root tips survival, or 200 root tips for RNA sample) and experimental repeats for all figures in the corresponding legends.

Minor suggestions:

Figure 4: although no mention of a potential role of RAP2.3 is made, and this has not been tested using genetic approaches, the authors use RAP2.3 in panel 4e to make their point for a connection between PGB1 overexpression, RAP2 levels and pre-adaptation to hypoxia. If similar blots could be conducted using RAP2.12 for example, this would strengthen the data further and would correlate with the genetic analysis at the beginning of the article.

Response: It would have been ideal to include crosses of *pgb1-1* & *35S:PGB1* with RAP2.12 protein reporter lines. Unfortunately, despite extensive efforts we were unable to successfully isolate homozygous F2 or F3 lines of these crosses. However, we think the evidence we show for the dependency of ethylene-mediated NO depletion for both RAP2.12 and RAP2.3 stability (in Figure 2b-c, Figure 3c-e and Supplementary Figure 7d) is convincing. Both RAP2.12 and RAP2.3 stability follow almost identical dynamics in response to all treatments and we think that RAP2.3 stability in PGB1 mutant backgrounds is therefore still relevant and convincing (also as a proxy for RAP2.12) (Figure 4), even though RAP2.3 alone is not clearly required for the enhanced tolerance.

In addition, to further strengthen our conclusions, we generated homozygous *pgb1-1* x RAP2.3-HA lines. Experiments with these lines provide more evidence that PGB1 is required for ERFVII stability in response to ethylene. These results are now included in Figure 4e.

Letters for figure panels are incorrect in some figure legends or on some figures (Figures 2 and 4).

Response: We corrected the letters in the Figure panels.

Figure 4b: can the authors comment on the presence of a lower molecular weight protein recognized by the anti-PGB1 antibodies in the *pgb-1* mutant background?

Response: We cannot state with certainty what exactly the lower MW bands represent. It is possible that it is a fragmented version of PGB1, or even other PGBs. We have now included a line in the text acknowledging this observation (Line 170-171, page 6). However, this does not influence our main conclusions as there is a clear absence of the ethylene responses in *pgb1-1* mutants, unlike the wild type plants.

Reviewer #2 (Remarks to the Author):

In this paper, the authors report that the gaseous phytohormone ethylene can upregulate PHYOGLOBIN-1 (PGB1), a nitric oxide (NO)-scavenger. The PGB1-mediated NO reduction stabilizes a transcription

factor called Group VII ETHYLENE RESPONSE FACTOR (ERFVII). Thus, when a plant first detects flooding, ethylene-triggered NO depletion and the subsequent increase in ERFVII protein help it to prepare for the hypoxic stress that is expected to follow. This study provides a nice example of how the adaptation of plants to flooding stress involves the coordinated signaling of three gases (O₂, ethylene and NO). Understanding the molecular mechanisms involved in this signaling is essential for the breeding of flood-tolerant crops. Abiotic stresses associated with climate change are likely to affect crop production. Thus, the mechanisms by which plants respond and adapt to abiotic stresses are of great interest. This study should therefore be of interest to many readers of Nature Communications.

I suggest that the authors consider the following points.

L50-52 (and Extended Data Fig. 1a, b & c): The authors evaluated how fast ethylene was perceived in root tips upon submergence, based on the intensities of fluorescence of the EIN3-GFP fusion protein. The authors need to explain how an increase in the fluorescence of EIN3-GFP is an indication that the root tips have perceived ethylene. Relevant papers should be cited for this.

Response: We agree that this was not clear in the text. This is now explained in the text together with relevant references (Line 60-63, page 3).

L56: "S2 & S3" should be "Extended Data Figs. 2 & 3".

Response: This figure reference was indeed incorrect. We have amended the figure reference according to the Nature Communications format throughout the manuscript.

L84-85 (and Fig. 2a): The authors state that ethylene-induced hypoxia tolerance redundantly involves RAP-type ERFVIIIs, which implies that RAP2.3 shares this redundancy. However, Fig. 2a shows that the double mutants *rap2.2/rap2.12* (in which RAP2.3 is still functional) do not have improved hypoxia tolerance after ethylene treatment. This seems to say that that RAP2.3 is not functionally redundant with RAP2.2 and RAP2.12 regarding hypoxia tolerance. The authors should explain their functions in detail.

Response: We have explained the role and involvement of constitutively expressed RAP-type ERFs more extensively in the main text (line 101-102, page 4; 140-141, page 5; 148-152, page 6). Here, we have also amended statements that implied a specific involvement of RAP2.3 in the ethylene-regulated response and clarify that we mostly use it as a proven reliable tool (Gibbs et al., *Nature* 2011, & *Molecular Cell*, 2014) to study post-translational regulation of ERFVIIIs by the PRT6 N-degron pathway.

L95: "Fig. S6b" should be "Extended Data Fig. 6b".

Response: We corrected this.

Fig. 2c: label "b" should be label "c".

Response: We corrected this.

Fig. 3d and 3f: The ethylene and ethylene+NO pre-treatments had similar effects on the intensities of RAP2.12::GFP fluorescence under subsequent hypoxia (Fig. 3d), but they had different effects on the rates of root tip survivals in Col-0 (Fig. 3f). A discussion about the different effects is needed.

Response: This discussion point is elaborated in the text (148-152, page 6). While ethylene and ethylene + NO pre-treated seedlings do indeed have similar behavior for RAP2.12 and RAP2.3 upon hypoxia (in Figure 3c-e and Supplementary Figure 7d), they differ during the pre-treatment due to differences in NO levels. We therefore drew the conclusion that ethylene-mediated enhanced root tip survival depends on enhanced ERFVII stability prior to hypoxia, and not during hypoxia.

Fig. 3e and Fig, 4e: The authors should measure RAP2.3-HA protein levels to examine the effects of NO and ethylene+NO pre-treatments by subsequent hypoxia treatment on accumulations of RAP2.3-HA protein. In this way, we can understand that the ethylene-mediated RAP2.3 stabilities are not abolished under subsequent hypoxia even if NO is pre-treated with ethylene, as like the case of RAP2.12::GFP fluorescence shown in Fig. 3d.

Response: We have performed this proposed experiment and added it to Supplementary Figure 7d and also described it in the main text (line 138-140, page 5). The results indicate that the absence of RAP2.12 and RAP2.3 stability during NO pre-treatments are due to enhanced N-degron proteolysis, as they can still stabilize under subsequent hypoxia.

Extended Data Fig. 3: It is difficult to compare the cell damage in root tips between the air and ethylene pre-treatments. Can the authors quantify the Evans blue-stained colors?

Response: We have now quantified the Evans blue staining in area and intensity and added 2 more time-points including the recovery phase (Supplementary Figure 3). These results indicate that ethylene promotes cell viability during both hypoxia and post-hypoxia.

Extended Data Fig. 4: There seems to be no common thread to the three panels (panel a, b and c show rosette FW, root tip survival and seedling survival, respectively). Is this figure needed?

Response: We believe Supplementary Figure 4 broadens the impact and relevance of our work, by showing that ethylene-induced hypoxia tolerance is conserved across various species and ecotypes, in addition to affecting both shoot and root tissues in response to both lethal and sub-lethal hypoxia durations. We agree this was not clear from the text and figure legends and have elaborated more on this (line 73-74, page 3).

Reviewer #3 (Remarks to the Author):

The manuscript (MS) submitted by Hartman et al. studies the cross road between gaseous molecules, and their regulators, that are known to be involved in plant responses to hypoxia stress. This new approach was intended to give answers on the participation of Ethylene (Ethy), nitric oxide (NO) and oxygen (O₂) in the regulation processes involved in the program that plants activate by low oxygen availability. Authors have studied the molecular mechanisms underpinning the adaptive process to hypoxia by which plants pretreated by low oxygen conditions are better prepared to respond to a subsequent hypoxic stress. In brief, they have mimicked hypoxia by pre-treating plants with Ethy, NO or low oxygen and studied how plants survive to a next and more severe situation of low availability of oxygen like flooding condition. These are classical priming experiments. Authors go deeper in the

understanding of plant responses to hypoxia trying to decipher the link between gases and some of their regulators. Knowing signaling pathways in responses to flooding stress could contribute to the design of crop plants more tolerant to the consequences of climate change.

Response to hypoxia is characterized by the stability of Group VII ethylene response factor (ERFVII), due to hypoxia-induced inhibition of the N-degron pathway, which is mediated by Ethylene and NO. Oxygen and NO oxidizes and destabilizes ERFVII and prevents responses to hypoxia. Hypoxia-induced Phytoalexins expression depletes increases of the NO concentration during hypoxia. Authors use mutants in the N-degron pathway, ERFVII factor, Phytoalexins (knock out mutants and overexpressed) to demonstrate the linkage between the gaseous molecules in the signaling pathway.

The experiments keep rationale and results are consistent with the statements. The novelty of results is barely at the standard level required by the journal.

The main points authors may address to improve the MS are the following.

1.- Title should be more specific. For instance, when authors say 'Gas signalling', clearly name the involved gaseous molecules that have been studied. Readers should clearly comprehend the statement. Do all studied gases allow plants pre-adapt to survive hypoxia stress? Which are those gases? In addition, title should clearly and specifically highlight the novelty of the attained results.

Response: We agree with the reviewer that the title was not very specific. We have now adjusted the title to more accurately reflect the message of the study:

Ethylene-mediated nitric oxide depletion pre-adapts plants to hypoxia stress

2.- It has been recently described that a hypoxic niche is required to set the activity of meristem cells through the regulation of the N-degron pathways, thus controlling the activity of stem cells (Weits et al., *Nature*, May-2019). In that report, authors conclude that hypoxic conditions were studied not as a limiting factor, but rather as a condition in shoot apical meristem cells to promote leaf organogenesis. Given that in the submitted MS, authors Hartman et al. used the root tip survival (where resides the root tip meristem) and root growth as a read out of plant response to hypoxic stress; it would be nice to integrate and discuss these results with those published in the recent *Nature* article, for instance integrating the roles of the presence or absence of oxygen in terms of both a signaling molecule and a critical substrate for cell metabolic activity.

Stem cells in root tip should keep in a hypoxic niche to commit their function, Are Ethylene, NO and phytoalexins present in root meristem and playing there a different function than they do on differentiated cells?, Are there different signaling pathways responding to hypoxia in the stem cells of meristem compared to other cells? How authors analyze the priming treatments with gas and their effects, when integrating all responses of target cells considering their dissimilar intrinsic characteristics?

Response: The work on hypoxic niches that the reviewer mentions, was not yet published at the time of the submission of this manuscript. We definitely agree that the recent Weits *et al.* (2019) paper published in *Nature* is very relevant and interesting and must be referred to in the current work. We have now discussed and cited this paper at multiple relevant stages throughout our manuscript (line 123-127 and 204-207, pages 5 and 7).

Previous studies investigating cell type specific responses to hypoxia (Mustroph *et al.*, *Plant Physiology*, 2010), have found that the core hypoxia response was conserved across all cell types. In

the current study we did not focus on cell type specific responses and therefore cannot comment on whether the mechanism of ethylene-mediated hypoxia priming would operate similarly across different cell types. This would be a very interesting avenue for future studies.

REVIEWERS' COMMENTS:

Reviewer #1 (Remarks to the Author):

The authors have addressed all my comments in a satisfactory manner and have made an effort to provide additional data requested.

Reviewer #2 (Remarks to the Author):

The revised manuscript has been improved in response to the reviewers' comments.

I found one minor point in Supplementary Figure 3b-c of the revised manuscript. The label "(Post-) hypoxia duration (h)" of x-axis of graphs in Supplementary Figure 3b-c should be shown separately by "Hypoxia duration (h)" for 2-4h hypoxia and "Post-hypoxia duration (h)" for 1-2h of recovery.

Response to Reviewers' comments after second round of peer-review:

We thank the reviewers for their feedback. We have adjusted the last remaining issue in the updated manuscript and figures.

Reviewer #1 (Remarks to the Author):

The authors have addressed all my comments in a satisfactory manner and have made an effort to provide additional data requested.

We thank reviewer #1 for her or his feedback.

Reviewer #2 (Remarks to the Author):

The revised manuscript has been improved in response to the reviewers' comments.

I found one minor point in Supplementary Figure 3b-c of the revised manuscript. The label "(Post-) hypoxia duration (h)" of x-axis of graphs in Supplementary Figure 3b-c should be shown separately by "Hypoxia duration (h)" for 2-4h hypoxia and "Post-hypoxia duration (h)" for 1-2h of recovery.

We thank reviewer #2 for her or his feedback and have corrected this issue in Supplementary Figure 3b&c.